# Uptake of Phosphate, Calcium, and Vitamin D by the Pregnant Uterus of Sheep in Late Gestation: Regulation by Chorionic Somatomammotropin Hormone

**DOI:** 10.3390/ijms23147795

**Published:** 2022-07-14

**Authors:** Claire Stenhouse, Katherine M. Halloran, Amelia R. Tanner, Larry J. Suva, Paul J. Rozance, Russell V. Anthony, Fuller W. Bazer

**Affiliations:** 1Department of Animal Science, Texas A&M University, College Station, TX 77843, USA; clairestenhouse@tamu.edu (C.S.); kittyhalloran14@tamu.edu (K.M.H.); 2College of Veterinary Medicine, Colorado State University, Fort Collins, CO 80523, USA; amelia.tanner@colostate.edu (A.R.T.); russ.anthony@colostate.edu (R.V.A.); 3Department of Veterinary Physiology and Pharmacology, Texas A&M University, College Station, TX 77843, USA; lsuva@cvm.tamu.edu; 4Department of Pediatrics, University of Colorado Anschutz Medical Campus, Aurora, CO 80045, USA; paul.rozance@cuanschutz.edu

**Keywords:** calcium, caruncle, chorionic somatomammotropin, cotyledon, phosphate, sheep, vitamin D

## Abstract

Minerals are required for the establishment and maintenance of pregnancy and regulation of fetal growth in mammals. Lentiviral-mediated RNA interference (RNAi) of chorionic somatomammotropin hormone (CSH) results in both an intrauterine growth restriction (IUGR) and a non-IUGR phenotype in sheep. This study determined the effects of CSH RNAi on the concentration and uptake of calcium, phosphate, and vitamin D, and the expression of candidate mRNAs known to mediate mineral signaling in caruncles (maternal component of placentome) and cotyledons (fetal component of placentome) on gestational day 132. CSH RNAi Non-IUGR pregnancies had a lower umbilical vein–umbilical artery calcium gradient (*p* < 0.05) and less cotyledonary calcium (*p* < 0.05) and phosphate (*p* < 0.05) compared to Control RNAi pregnancies. CSH RNAi IUGR pregnancies had less umbilical calcium uptake (*p* < 0.05), lower uterine arterial and venous concentrations of 25(OH)D (*p* < 0.05), and trends for lower umbilical 25(OH)D uptake (*p* = 0.059) compared to Control RNAi pregnancies. Furthermore, CSH RNAi IUGR pregnancies had decreased umbilical uptake of calcium (*p* < 0.05), less uterine venous 25(OH)D (vitamin D metabolite; *p* = 0.055), lower caruncular expression of *SLC20A2* (sodium-dependent phosphate transporter; *p* < 0.05) mRNA, and lower cotyledonary expression of *KL* (klotho; *p* < 0.01), *FGFR1* (fibroblast growth factor receptor 1; *p* < 0.05), *FGFR2* (*p* < 0.05), and *TRPV6* (transient receptor potential vanilloid member 6; *p* < 0.05) mRNAs compared to CSH RNAi Non-IUGR pregnancies. This study has provided novel insights into the regulatory role of CSH for calcium, phosphate, and vitamin D utilization in late gestation.

## 1. Introduction

Across vertebrate species, a constant and extensive supply of minerals is required to ensure the formation and maintenance of a strong, durable, dynamic, and renewable skeleton [1]. Phosphorous and calcium, two of the most abundant minerals in the body, have critical roles in the development and post-natal function of both the kidney and skeleton. Additionally, these mineral ions have important roles in the regulation of many physiological processes including cellular metabolism, proliferation, and protein synthesis [1,2,3,4,5,6,7]. 

Postnatally, mineral and bone homeostasis are tightly regulated by interactions of parathyroid hormone (PTH), PTH related protein (PTHrP), 1,25-dihydroxyvitamin D3 (1,25[OH]2D3), fibroblast growth factor 23 (FGF23)-klotho (KL) signaling with FGF receptors, and sex steroids (estradiol, progesterone, and testosterone) [1,8,9]. These hormones regulate mineral metabolism via the expression of regulatory molecules such as calcium-transporting ATPases, calcium-binding proteins, sodium dependent phosphate transporters, and the transient receptor potential vanilloid (TRPV) family members [6,10,11,12,13,14,15]. Emerging evidence suggests that many of these molecules with important roles in the regulation of mineral transport postnatally are expressed at the ovine maternal-conceptus interface throughout gestation [16,17]. As such, these molecules regulate the active transport of minerals from the maternal circulation to the fetal-placental circulation where they are essential for the appropriate regulation of fetal development [1,18].

It has been suggested that interferon tau (IFNT), the maternal signal for pregnancy recognition in ruminants, and progesterone (P4) act both independently and synergistically to alter endometrial expression of mRNAs and proteins important for calcium, phosphate, and vitamin D signaling, transport, and utilization during the peri-implantation period of pregnancy [19]. Furthermore, it was recently reported that exogenous P4 supplementation for the first eight days of pregnancy altered expression of mRNAs and proteins of regulators of calcium, phosphate, and vitamin D signaling, transport, and utilization in both the endometrium and placentomes on Day 125 of pregnancy [20]. While these studies have provided compelling evidence for endocrine (P4) and paracrine (IFNT) regulation of mineral transport in the pregnant ewe, other conceptus (embryo/fetus and associated extra-embryonic membranes) derived signals may be important regulators of phosphate, calcium, and vitamin D signaling, transport, and utilization at the maternal-conceptus interface. 

The binucleate cells of the ovine conceptus begin secreting chorionic somatomammotropin hormone (CSH; also known as placental lactogen) into the uterine lumen from approximately Day 16 of gestation [21]. CSH continues to be expressed by the binucleate cells and the syncytium of the ovine placentomes throughout gestation [22] and is detectable in the maternal uterine vein from approximately Day 45 of gestation and in peripheral serum from approximately Day 48 of gestation [23]. Peak concentrations of CSH in peripheral serum are between Days 120 to 140 of gestation, before rapidly declining approximately five days prior to parturition [23]. 

Recent studies utilizing lentiviral-mediated RNA interference (RNAi) in pregnant sheep have demonstrated an important role of CSH in the regulation of fetal growth. The knockdown of CSH protein led to the formation of two fetal phenotypes: non-intrauterine growth restricted and intrauterine growth restricted (IUGR) [24,25]. In non-IUGR pregnancies, CSH RNAi reduced uterine blood flow and the expression of endothelial nitric oxide synthase (NOS3) protein in caruncles, while increasing utero-placental glucose utilization [25]. In contrast, in CSH RNAi IUGR pregnancies, reduced uterine and umbilical blood flow, utero-placental glucose utilization, and uterine uptake of amino acids was accompanied by lighter uterine and fetal weights [24]. These findings highlight the potential role of CSH in the regulation of nutrient transport in the pregnant sheep.

The relative abundance of phosphate and calcium in ovine fetal fluids (allantoic and amniotic fluids) in late gestation have been reported [16,17]. However, the concentration of phosphate, calcium, and vitamin D in uterine and umbilical vessels and how these nutrients are utilized by the fetus, placental membranes, and uterus are not well documented. Given the previously described effects of CSH RNAi on the regulation of fetal growth, uterine and umbilical blood flow, and nutrient transport, it was hypothesized that CSH RNAi would decrease the utero-placental utilization of phosphate, calcium, and vitamin D, with a more severe phenotype observed in IUGR pregnancies. This study aimed to determine concentrations of phosphate, calcium, and the vitamin D metabolite 25(OH)D in uterine and umbilical blood vessels. Additionally, the study was designed to determine the impact of CSH RNAi on concentrations of phosphate, calcium, and 25(OH)D in uterine vessels, caruncles, and cotyledons, as well as the expression of candidate mRNAs with known roles in the regulation of phosphate, calcium, and vitamin D in caruncles and cotyledons in both Non-IUGR and IUGR pregnancies. 

## 2. Results

### 2.1. Concentrations of Phosphate, Calcium, and 25(OH)D in Uterine and Umbilical Blood 

Plasma samples from Control RNAi pregnancies on Day 132 of gestation were used to determine the concentrations of phosphate, calcium, and 25(OH)D in uterine and umbilical vessels in late gestation. Calcium was more abundant in umbilical blood vessels than uterine blood vessels (Figure 1A; Overall vessel effect *p* < 0.0001). Phosphate tended to be more abundant in umbilical arteries than uterine veins on Day 132 of gestation (post-hoc comparison *p* = 0.056); Figure 1B). The concentrations of 25(OH)D were not different in plasma from each of the vessel types investigated (Figure 1C). 

### 2.2. The Effect of CSH RNAi on Calcium, Phosphate, and Vitamin D in Plasma 

#### 2.2.1. Calcium

The uterine uptake of calcium was less in CSH RNAi IUGR pregnancies compared to CSH RNAi Non-IUGR pregnancies (*p* < 0.05; Table 1). Similarly, the umbilical uptake of calcium was less in CSH RNAi IUGR pregnancies compared to Control RNAi pregnancies (*p* < 0.05; Table 1. Overall treatment effect *p* = 0.089). Furthermore, CSH RNAi Non-IUGR pregnancies had a lower umbilical vein–umbilical artery calcium gradient than Control RNAi pregnancies (*p* < 0.05; Table 1). CSH RNAi did not affect concentrations of calcium in plasma from uterine arteries or umbilical vessels, the uterine artery–vein calcium gradient, the uteroplacental utilization of calcium, or the uterine and umbilical uptakes of calcium relative to uterine, fetal and placental weights (Table 1).

#### 2.2.2. Phosphate

It is important to note that plasma from uterine arterial samples from non-IUGR pregnancies was unavailable for phosphate analyses. There were no significant differences in concentrations of phosphate in uterine or umbilical vessels, uterine artery-vein or umbilical vein-artery gradients, uterine or umbilical uptakes, or uteroplacental utilization of phosphate between Control RNAi and CSH RNAi IUGR pregnancies (Appendix A).

#### 2.2.3. 25(OH)D

It is important to note that plasma from uterine arterial samples from non-IUGR pregnancies was unavailable for 25(OH)D analyses. 25(OH)D was less abundant in plasma from uterine arteries (*p* < 0.05; Table 2) and veins (*p* < 0.05; Table 2) from CSH RNAi IUGR pregnancies compared to Control RNAi pregnancies. Similarly, 25(OH)D tended to be less abundant in plasma from uterine veins from CSH RNAi IUGR pregnancies compared to CSH RNAi Non-IUGR pregnancies (*p* = 0.055; Table 2). Furthermore, there tended to be an effect of CSH RNAi on the umbilical uptake of 25(OH)D per kg of fetus (Overall Treatment Effect *p* = 0.09), with less 25(OH)D uptake in the CSH RNAi IUGR pregnancies compared to Control RNAi pregnancies (posthoc comparison *p* = 0.059; Table 2). There were no significant differences in concentrations of 25(OH)D in umbilical vessels, uterine artery-vein or umbilical vein-artery gradients, uterine or umbilical uptakes, or uteroplacental utilization of 25(OH)D between Control RNAi and CSH RNAi IUGR pregnancies (Table 2).

### 2.3. Effect of CSH RNAi on Abundances of Calcium and Phosphate in Cotyledons and Caruncles

Caruncles from CSH RNAi Non-IUGR pregnancies tended to have less calcium than CSH RNAi IUGR pregnancies (*p* = 0.053; Figure 2A). Similarly, calcium (*p* < 0.05; Figure 2B) and phosphate (*p* < 0.05; Figure 2D) were less abundant in cotyledons from CSH RNAi Non-IUGR pregnancies compared to Control RNAi pregnancies. CSH RNAi did not alter abundance of phosphate in caruncles (Figure 2C) on Day 132 of gestation.

### 2.4. Effect of CSH RNAi on the Expression of Candidate mRNAs with Roles in the Regulation of Calcium, Phosphate, and Vitamin D Signaling in Caruncles and Cotyledons

The expression of the sodium dependent phosphate transporter (solute carrier family 20 member 2) *SLC20A2* mRNA was lower in caruncles from CSH RNAi IUGR pregnancies compared to CSH RNAi Non-IUGR pregnancies (posthoc comparison *p* < 0.05; overall treatment effect *p* = 0.056. Appendix A). CSH RNAi did not affect the expression of *KL*, *FGFR1*, *FGFR2*, *ADAM10*, *ADAM17*, *PTHRP*, *SLC20A1*, *STC1*, *STC2*, *ATP2B4*, *PTHRP*, *S100G*, *TRPV6*, *CYP24*, or *VDR* mRNAs in caruncles (*p* > 0.10; Appendix A). There were treatment effects of CSH RNAi on the expression of *KL* (*p* = 0.098; Figure 3A), *FGFR1* (*p* < 0.05; Figure 3B), *FGFR2* (*p* < 0.05; Figure 3C), *S100A9* (*p* = 0.067; Figure 3J), *STC2* (*p* = 0.067; Figure 3L) and *CYP24* (*p* = 0.096; Figure 3N) mRNAs in cotyledons. Cotyledons from CSH RNAi IUGR pregnancies had lower expression of *CYP24* (*p* < 0.05; Figure 3N) mRNA compared to Control RNAi pregnancies. Similarly, the expression of *KL* (*p* = 0.069; Figure 3A), *FGFR1* (*p* = 0.09; Figure 3B), and *STC2* (*p* = 0.059; Figure 3L) mRNAs tended to be lower in cotyledons from CSH RNAi IUGR pregnancies compared to Control RNAi pregnancies. Furthermore, the expression of *KL* (*p* < 0.01; Figure 3A), *FGFR1* (*p* < 0.05; Figure 3B), *FGFR2* (*p* < 0.05; Figure 3C), *S100A9* (*p* = 0.057; Figure 3J), *STC2* (*p* = 0.089; Figure 3L), *TRPV6* (*p* < 0.05; Figure 3M), and *CYP24* (*p* = 0.06; Figure 3N) mRNAs was lower in cotyledons from CSH RNAi IUGR pregnancies compared to CSH RNAi Non-IUGR pregnancies. CSH RNAi did not affect the expression of *ADAM10*, *ADAM17*, *SLC20A1*, *SLC20A2*, *STC1*, *ATP2B4*, *S100G*, *TRPV6*, or *VDR* mRNAs in cotyledons (*p* > 0.10; Figure 3).

### 2.5. Correlations between Biometric Data and Caruncular and Cotyledonary Phosphate and Calcium Abundance

To further assess the relationship between the abundance of calcium and phosphate and biometric data from this study, maternal, uterine, fetal, and placental weights were correlated with the abundance of phosphate and calcium in caruncular and cotyledonary homogenates (Appendix A). The abundance of calcium in caruncles tended to be negatively associated with fetal weight (r = 0.181; *p* = 0.06). In contrast, the abundance of both calcium (r = 0.163; *p* = 0.09) and phosphate (r = 0.365; *p* < 0.01) were positively correlated to placental weight. 

### 2.6. Correlations between Biometric Data and Calcium, Phosphate, and Vitamin D Utilization

Maternal and uterine (Appendix A), and fetal and placental (Appendix A) weights were correlated against the abundance, uterine uptake, umbilical uptake, uterine artery-umbilical artery gradient, uterine artery-uterine vein gradient, and uteroplacental utilization of phosphate, calcium, and 25(OH)D. The uteroplacental utilization of phosphate tended to be positively correlated with fetal weight (r = 0.483; *p* = 0.056). The abundance of calcium in umbilical arterial (r = 0.161; *p* = 0.08) and venous (r = 0.181; *p* = 0.06) plasma tended to be positively correlated with placental weight. Furthermore, umbilical calcium uptake was positively correlated with uterine weight (r = 0.174; *p* = 0.067), fetal weight (r = 0.499; *p* < 0.001), and placental weight (r = 0.319; *p* < 0.01). The abundance of 25(OH)D in uterine venous plasma was positively correlated with fetal weight (r = 0.294; *p* < 0.05). In contrast, the uterine artery-uterine vein calcium gradient was negatively correlated with uterine weight (r = 0.276; *p* < 0.05), fetal weight (r = 0.334; *p* < 0.01), and placental weight (r = 0.210; *p* < 0.05).

### 2.7. Correlations between Biometric Data and the Caruncular and Cotyledonary mRNA Expression of Calcium, Phosphate, and Vitamin D Regulatory Molecules

Maternal, uterine, fetal, and placental weights were correlated against the caruncular and cotyledonary expression of calcium, phosphate, and vitamin D regulatory mRNAs (Table 3). Maternal weight was negatively correlated with the caruncular expression of PTHrP (r = 0.262; *p* < 0.05) and STC1 (r = 0.233; *p* < 0.05) mRNAs. Uterine weight was negatively correlated with the caruncular expression of ADAM10 (r = 0.226; *p* < 0.05). Likewise, ATP2B4 (r = 0.185; *p* = 0.058), PTHrP (r = 0.177; *p* = 0.072), and STC1 (r = 0.199; *p* = 0.06) mRNAs tended to be negatively correlated with uterine weight. Furthermore, fetal weight was negatively correlated with the caruncular expression of ADAM10 (r = 0.161; *p* = 0.079), SLC20A1 (r = 0.241; *p* < 0.05), and PTHrP (r = 0.281; *p* < 0.05) mRNAs. Similarly, the caruncular expression of ADAM10 (r = 0.159; *p* = 0.08) and SLC20A1 (r = 0.288; *p* < 0.05) mRNAs was negatively correlated with placental weight. In contrast, the expression of SLC20A2 mRNA by caruncles tended to be positively correlated with fetal weight (r = 0.167; *p* = 0.074). The cotyledonary expression of VDR mRNA was negatively correlated with fetal (r = 0.224; *p* < 0.05) and placental (r = 0.208; *p* < 0.05) weight. Maternal weight was positively correlated with the cotyledonary expression of FGFR2 mRNA (r = 0.389; *p* < 0.01) and tended to be positively correlated with ADAM10 (r = 0.158; *p* = 0.083), and SLC20A1 (r = 0.195; *p* = 0.059) mRNAs. Similarly, cotyledonary KL mRNA expression tended to be correlated with uterine weight (r = 0.184; *p* = 0.067). Furthermore, fetal weight was positively correlated with the cotyledonary expression of KL (r = 0.222; *p* < 0.05), FGFR1 (r = 0.194; *p* = 0.05), and S100A9 (r = 0.225; *p* < 0.05) mRNAs.

## 3. Discussion

Findings from studies across eutherian mammals suggest a critical role for the utero-placental transport of minerals for the establishment and maintenance of pregnancy and for the regulation of fetal-placental growth and development [1,18]. Despite the wide acceptance that phosphate, calcium, and vitamin D are essential during pregnancy, few details exist regarding the hormonal regulation of mechanisms required for transport and metabolism of these fundamental nutrients. 

While several techniques exist to study placental mineral transport in pregnant ewes [1], the most widely accepted is in situ placental perfusion [26,27]. However, although this technique has provided significant enhancements in the understanding of mineral transport in the pregnant sheep, it requires removal of the fetus that leads to a rapid decline in placental function. Thus, caution should be taken when extrapolating the significance of these findings. In contrast, the present study utilized catheters which did not disrupt placental or fetal functions and, therefore, provides important new advances regarding the understanding of mineral transport, uptake, and utilization by the uterus and conceptus in late gestation. The utilization of real-time, steady-state quantification of the abundance of calcium in relation to uterine and umbilical blood flow provides a reliable assessment of calcium flow through the feto-placental unit. Phosphate, calcium, and vitamin D signaling pathways are intimately interrelated [18] and, therefore, placental transport and utilization of these nutrients must not be considered in isolation. Our real-time calcium measurements, in combination with the quantification of phosphate and 25(OH)D in uterine and umbilical samples by colorimetric methods and ELISA, respectively, provide important insights into mineral transport during late gestation in pregnant ewes. 

It is widely appreciated that mammalian fetuses are hypercalcemic compared to maternal calcium levels in late gestation, a process ensuring appropriate skeletal mineralization [1]. Considering this, it was unsurprising to find a greater abundance of calcium in umbilical vessels (both artery and vein) compared to uterine vessels (both artery and vein) in control animals in the present study. This is consistent with extensive transport of calcium to the fetus, as demonstrated in Table 1, which was expected as substantial amounts of calcium accumulate in both amniotic and allantoic fluid for utilization by the fetus in late gestation [17]. Further, placental transport of minerals is primarily uni-directional, ensuring that adequate mineral supplies are available for the fetus. Thus, the back-flux of calcium from conceptus to mother is estimated to be as little as 0.4% in sheep in late gestation [28]. 

Prolactin and CSH have been suggested to regulate intestinal calcium absorption, reduce urinary calcium excretion [29], and stimulate synthesis of PTHrP [30]. Regulation of intestinal mineral absorption is a critical adaptation to increase the availability of minerals from the maternal circulation [18], thereby preventing hypocalcemia during periods of high demands of the fetus for minerals, particularly during late gestation. Intrauterine infusions of ovine CSH increase the abundance of STC1 in progestinized ewes [31], demonstrating important roles for P4 and CSH in the regulation of calcium transport by the ovine uterus. However, to date, there have been limited investigations into the role of CSH in the regulation of calcium homeostasis in late gestation, when there is abundant expression of CSH by binucleate cells and the syncytium of ovine placentomes and its release into the maternal peripheral circulation [22,23,32]. The present study provides novel evidence for a role of CSH in the regulation of calcium homeostasis during late gestation in sheep. CSH RNAi IUGR pregnancies had less uterine calcium uptake than Control RNAi pregnancies. Similarly, CSH RNAi IUGR pregnancies had less umbilical calcium uptakes and lower cotyledonary expression of *TRPV6* mRNA than CSH RNAi Non-IUGR pregnancies. Interestingly, calcium homeostasis also appeared to be altered in the CSH RNAi Non-IUGR pregnancies. The umbilical vein-umbilical artery calcium gradient was lower in CSH RNAi Non-IUGR pregnancies, suggesting either lower utilization or greater availability of calcium for CSH RNAi Non-IUGR fetuses compared to both the Control RNAi and CSH RNAi Non-IUGR fetuses. Additionally, results of this study revealed a lower abundance of calcium in the cotyledons of CSH RNAi Non-IUGR pregnancies when compared to Control RNAi pregnancies. Although these fetuses were not considered to be intrauterine growth restricted, CSH RNAi does reduce uterine blood flow, decreases the abundance of NOS3 protein, and increases the utero-placental utilization of glucose compared to Control RNAi pregnancies [25]. Clearly, despite no gross differences in fetal weight, there are nutritional and potentially metabolic perturbations in the CSH RNAi Non-IUGR pregnancies which warrant further investigation. 

Interestingly, the culture of human term trophoblast cells with the calcium ionophore A23187 inhibited the secretion of CSH [33], suggesting that calcium exerts a negative effect on the release of CSH from term human trophoblast cells. A relationship between calcium and CSH release in vitro has also been suggested in the sheep [34]. Ovine cotyledonary cells release greater amounts of CSH in response to culture with (1) calcium-depleted medium, (2) the calcium antagonist MgCl, and (3) the calcium channel blocking agents verapamil and nefidipine, suggesting an inverse relationship between extracellular concentrations of calcium and CSH secretion. 

This study, to our knowledge, is the first to report a relationship between CSH and phosphate signaling and transport at the ovine maternal-conceptus interface. In the present study, there was less phosphate in cotyledons of CSH RNAi Non-IUGR pregnancies compared to Control RNAi pregnancies. Furthermore, the expression of sodium dependent phosphate transporter *SLC20A2* mRNA was lower in caruncles from CSH RNAi IUGR compared to CSH RNAi Non-IUGR pregnancies, and this was accompanied by the lower expression of mRNAs for components of the KL-FGF signaling cascade (*KL*, *FGFR1*, and *FGFR2*) in cotyledons from CSH RNAi IUGR pregnancies compared to Control RNAi and/or CSH RNAi Non-IUGR pregnancies. Given the intimate relationship between phosphate and calcium homeostasis [35], and the essential role of CSH in the regulation of placental nutrient transport, fetal growth, and development [36], it is perhaps unsurprising that alterations in phosphate signaling were observed in this study. As there were no effects of CSH RNAi treatment on abundance in uterine and umbilical vessels, uterine or umbilical uptake or utero-placental utilization of phosphate, local alterations in phosphate signaling may occur within placentomes to counteract the effects of CSH deficiency on calcium and vitamin D abundance. Further studies are required to fully understand the relationship between phosphate, calcium, vitamin D, and CSH at the maternal conceptus in sheep throughout gestation. 

In sheep it is known that radiolabeled 1,25-dihydroxycholecalciferol can be transported across the placenta in both directions [37], and that ovine fetuses have greater metabolism of vitamin D compared to maternal vitamin D metabolism [38]. It has been suggested that, in pregnant ewes, the placenta is responsible for the generation of approximately 40% of the 1,25-dihydroxycholecalciferol in fetal blood and that local metabolism of vitamin D in the pregnant uterus is critical for the regulation of fetal growth. In fact, the cytochrome P450 (CYP) enzymes critical for the metabolism of vitamin D, and the vitamin D receptor, are expressed at the maternal-conceptus interface in late gestation (Day 125) in sheep [17,20]. This finding is not unique to sheep as the expression of vitamin D regulatory molecules in the female reproductive tract in pigs, humans, and rodents has been reported [39,40,41,42], suggesting important roles of vitamin D in the establishment and maintenance of pregnancy, and in the regulation of conceptus growth and development.

In the present study, there were lower concentrations of 25(OH)D in uterine arterial and venous blood from CSH RNAi IUGR pregnancies, suggesting a systemic effect of CSH RNAi on maternal vitamin D status in pregnancies with IUGR fetuses. Human CSH has been reported to stimulate the activity of 1α-hydroxylase (CYP27B1) [43], an important enzyme for the generation of ‘active’ vitamin D in vivo. Furthermore, CSH can increase plasma 1,25-dihydroxyvitamin D in hypophysectomized, non-pregnant rats [44]. Collectively, these findings suggest that CSH may act upon the liver and kidneys to alter the metabolism of vitamin D. Thus, it could be speculated that CSH RNAi that resulted in IUGR pregnancies may have systemic effects on maternal vitamin D status that must be elucidated in future studies. The umbilical uptake of 25(OH)D was negative across treatment groups, indicating production of 25(OH)D by the fetus. Furthermore, the decreased expression of *CYP24* mRNA was observed in the cotyledons of CSH RNAi IUGR pregnancies compared to Control RNAi pregnancies. The activity of vitamin D is regulated by the catabolic activity of CYP24A1 (24-hydroxylase) which inactivates 1,25[OH]_2_D_3_ via conversion to 1,24,25[OH]_3_D_3_ [45]. Collectively, these findings suggest (1) alterations in maternal vitamin D status in response to CSH RNAi in ewes which have IUGR fetuses; and (2) that conceptuses (embryo/fetus and associated extra-embryonic membranes) from CSH RNAi IUGR pregnancies may increase the metabolism of vitamin D to compensate for the decrease in available maternal vitamin D. As these differences were unique to pregnancies with IUGR fetuses, they provide additional evidence for an important role of vitamin D in the regulation of fetal growth and suggest that CSH influences vitamin D homeostasis in the pregnant ewe. 

It is important to note that the majority of differences in the tissues in response to CSH RNAi were observed in the cotyledon and not the caruncle. This is perhaps unsurprising given that the CSH is expressed by the binucleate cells and the syncytium of the ovine placentomes [21,22]. Thus, the findings presented in this study support the concept that CSH regulates mineral transport in the fetal portion of the placentome.

## 4. Materials and Methods

### 4.1. Experimental Animals and Sample Collection

The experimental design and sample collection methods have been published with regard to the impact of chorionic somatomammotropin RNA interference (CSH RNAi) on uterine and umbilical blood flows, and nutrient utilization for lentiviral scrambled control RNAi (control, *n* = 10), normal-weight (non-IUGR, *n* = 6) and growth restricted sheep fetuses (IUGR, *n* = 4) [24,25]. In brief, all ewes (Dorper breed composition) were group housed in pens at the Colorado State University Animal Reproduction and Biotechnology Laboratory, and were provided access to hay, trace minerals, and water in order to meet or slightly exceed their National Research Council requirements. Following synchronization and subsequent breeding, fully expanded and hatched blastocysts were collected by flushing the uteri on Day nine of pregnancy. Each blastocyst was infected with 100,000 transducing units of either control RNAi or CSH RNAi virus. After infection for approximately 5 h, each blastocyst was washed, and a single blastocyst was transferred surgically into a synchronized recipient ewe. Each recipient ewe was monitored daily for return to standing estrus and confirmed pregnant on Day 50 of gestation by ultrasound (Mindray Medical Equipment, Mahwah, NJ, USA).

### 4.2. Quantification of Calcium, Phosphate, and 25(OH)D in Uterine Flushings, Plasma, and Endometrial Homogenates

As described previously, on Day 126 of gestation, ewes underwent surgical placement of fetal and maternal catheters in order to determine blood flow and nutrient flux. Briefly, the following catheters were placed: fetal descending aorta (representing umbilical artery blood), fetal femoral vein and umbilical vein, maternal femoral artery (representing uterine artery blood), maternal femoral vein, and uterine vein [25]. On Day 132 of gestation, uterine and umbilical blood flows were determined using the steady state ^3^H_2_O transplacental diffusion technique. Using these catheters, four samples of blood were obtained on Day 132 at 20-min intervals for the real time measurement of calcium (Ca^2+^) using a blood gas analyzer (ABL800 Flex, Radiometer, Brea, CA, USA). Day 132 was selected as peak concentrations of CSH in peripheral serum are found between Days 120 to 140 of gestation [23]. 

Snap-frozen samples from caruncles (maternal component of placentome) and cotyledons (fetal component of placentomes) (300–500 mg of tissue) were homogenized in 1 mL of lysis buffer (60 mM Tris-HCl (Sigma Aldrich, St. Louis, MO, USA), 1 mM Na_3_VO_4_ (Fisher Scientific, Waltham, MA, USA), 10% glycerol (Fisher Scientific), and 1% sodium dodecyl sulfate (BioRad, Hercules, CA, USA), containing an EDTA-free protease inhibitor (Roche, Indianapolis, IN, USA)). Homogenates were centrifuged at 14,000× *g* for 15 min at 4 °C, and supernatants were stored at −80 °C until assayed. The concentration of total proteins in the tissue homogenates was quantified spectrophotometrically (SynergyH1, BioTek, Winooski, VT, USA. Absorbance 562 nm) using a protein assay dye reagent (BioRad; 500-0006) according to the manufacturer’s instructions.

The concentrations of calcium in caruncular and cotyledonary homogenates were quantified using a colorimetric assay (Sigma Aldrich; MAK022), as described previously [17]. The concentrations of phosphate in uterine and umbilical plasma samples, and caruncular and cotyledonary homogenates were quantified using a colorimetric assay (Abcam, Cambridge, MA, USA; ab65622), as described previously [16]. Additionally, the concentrations of the vitamin D metabolite 25(OH)D in uterine and umbilical plasma samples were quantified by ELISA (Eagle Biosciences, Amherst, NH, USA; VID91-K01), as described previously [17].

Uterine and umbilical gradients, and uptakes of calcium, phosphate, and 25(OH)D relative to uterine, fetal, and placental weights were calculated using mathematical formulas published previously [25] (see Table 4). It is important to note that plasma from uterine arterial samples from non-IUGR pregnancies was unavailable for phosphate and 25(OH)D analyses. The abundance of calcium and phosphate in caruncular and cotyledonary homogenates are expressed relative to concentration of protein in each sample.

### 4.3. Analysis of Candidate Gene Expression by qPCR

Total RNA was extracted from ovine caruncles and cotyledons using Trizol (15596018 Invitrogen, Carlsbad, CA, USA) and further purified using the RNeasy Mini Kit (74104 Qiagen, Hilden, Germany) as described previously [17]. Concentrations of RNA were quantified using a Nanodrop (ND-1000 Spectrophotometer), and all samples had a 260/280 value >2. Complementary DNA (cDNA) was synthesized from 1 µg of RNA with SuperScript II reverse transcriptase and oligo (deoxythymidine) primers (Invitrogen, Carlsbad, CA, USA), as per the manufacturer’s instructions. Negative controls without reverse transcriptase were included to test for genomic contamination and all cDNA was stored at −20 °C until required.

The relative expression of candidate genes in cotyledonary and caruncular samples was quantified by quantitative polymerase chain reaction (qPCR) as described previously [16,17]. To investigate the effects of CSH on phosphate signaling in the sheep placentome, the mRNA expression of the sodium dependent phosphate transporters solute carrier family 20 member 1 (*SLC20A1*) and *SLC20A2*, and components of the fibroblast growth factor-klotho signalling cascade (fibroblast growth factor receptor 1 [*FGFR1*], *FGFR2*, klotho [*KL*], a disintegrin and metalloproteinase domain-containing protein 10 [*ADAM10*], and *ADAM17*) was quantified by qPCR. To investigate the effects of CSH on calcium signalling in sheep placentomes, the mRNA expression of the calcium binding proteins S100 calcium-binding protein G (*S100G*) and *S100A9*, plasma membrane calcium-transporting ATPase 4 (*ATP2B4*), PTH related protein (*PTHrP*), stanniocalcin 1 (*STC1*), *STC2*, and transient receptor potential cation channel subfamily v member 6 (*TRPV6*) was quantified by qPCR. Additionally, the expression of the 25-hydroxyvitamin D3-24-hydroxylase (*CYP24*), which inactivates 1,25[OH]2D3 via conversion to 1,24,25[OH]3D3, and vitamin D receptor (*VDR*) was quantified by qPCR. qPCR was performed using the ABI prism 7900HT system (Applied Biosystems, Foster City, CA, USA) with Power SYBR Green PCR Master Mix (Applied Biosystems), as per the manufacturer’s instructions to determine expression of mRNAs encoding for genes of interest. Primer sequences are provided in Appendix A. Primer efficiency and specificity were evaluated by generating a standard curve from pooled cDNA and by the inclusion of a dissociation curve to the RT reaction, respectively. Serial dilutions of pooled cDNA in nuclease-free water ranging from 1:2 to 1:256 were used as standards. All primer sets used amplified a single product and had an efficiency of between 95% and 105%. Each well contained 10% cDNA, 30% nuclease-free water, 10% primer, and 50% SYBR Green reaction mix in a 10 µL reaction volume. All reactions were performed at an annealing temperature of 60 °C. The stability of reference genes was assessed by geNORM V3.5 (Ghent University Hospital, Centre for Medical Genetics, Ghent, Belgium) in each tissue, with an M value of <1.5, and the reference genes used for normalization had stable expression. The reference genes *GAPDH* and *TUB,* and *ACTB* and *SDHA* were determined to have stable expression in caruncles and cotyledons, respectively, by geNORM V3.5 (Ghent University Hospital, Centre for Medical Genetics). Additionally, the effects of CSH RNAi were determined using GenStat (Version 13.1; VSN International Ltd., Indore, India) to ensure that there was no effect of CSH knockdown on reference gene expression. The abundances of the candidate mRNAs in the samples were quantified using the ΔΔCq method.

### 4.4. Statistical Analysis

All statistical analyses were performed using Minitab (Minitab, LLC, Chicago, IL, USA) or Graphpad Prism 9 Software (Graphstats Technologies Private Limited, Karnataka, India). Mean values were calculated for each individual sample for each parameter investigated and the normality of the distribution of the data was assessed using the Anderson-Darling test. If *p* < 0.05, the data were not considered to have a normal distribution. Transformations were carried out if necessary to achieve a Gaussian distribution. Outliers identified by a ROUT outlier test were excluded. ANOVA with a Tukey post-hoc analysis or t-tests were performed where appropriate to assess the effects of CSH RNAi on the parameters investigated. The effect of CSH RNAi on all parameters investigated was compared between (1) control RNAi, (2) CSH RNAi Non-IUGR, and (3) CSH RNAi IUGR groups. The results were considered significant at *p* < 0.05, trending towards significant at *p* > 0.05 < 0.1 and not significant at *p* > 0.1. Correlations were performed between maternal, uterine, fetal, and placental weights and all parameters investigated in this study. 

## 5. Conclusions

This study has provided novel insights into the abundance and relative utilization of calcium, phosphate, and vitamin D by ovine uteri and conceptuses in late gestation. The results provide evidence supporting a role for CSH in the regulation of mineral transport and utilization at the ovine maternal-conceptus interface, thus providing a platform upon which further mechanistic understanding of the hormonal regulation of mineral transport during pregnancy in mammalian species can be achieved. 

## Figures and Tables

**Figure 1 ijms-23-07795-f001:**
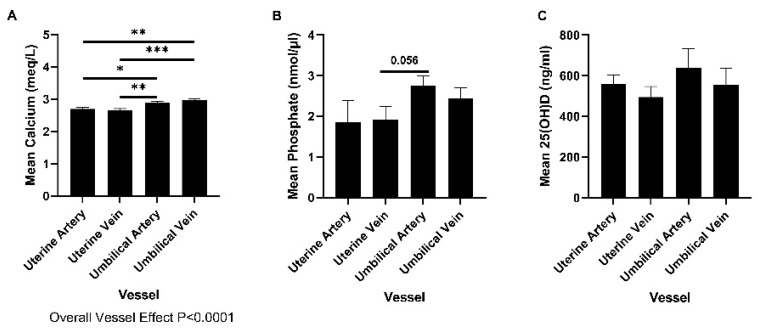
Concentration of calcium (**A**), phosphate (**B**), and 25(OH)D (**C**) in plasma from control RNAi uterine and umbilical blood collected on Day 132 of gestation. Data are presented as means ± SEM. Control RNAi, *n* = 10. * *p* < 0.05, ** *p* < 0.01, *** *p* < 0.001.

**Figure 2 ijms-23-07795-f002:**
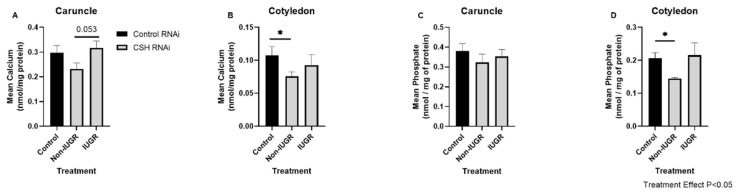
Abundances of calcium (**A**,**B**) and phosphate (**C**,**D**) in caruncles (**A**,**C**) and cotyledons (**B**,**D**) from Control RNAi, CSH RNAi Non-IUGR, and CSH RNAi IUGR pregnancies. Data are presented as means ± SEM. Control RNAi, *n* = 10; CSH RNAi Non-IUGR, *n* = 6; CSH RNAi IUGR, *n* = 4. CSH, chorionic somatomammotropin; IUGR, intrauterine growth restriction; RNAi, RNA interference. * *p* < 0.05.

**Figure 3 ijms-23-07795-f003:**
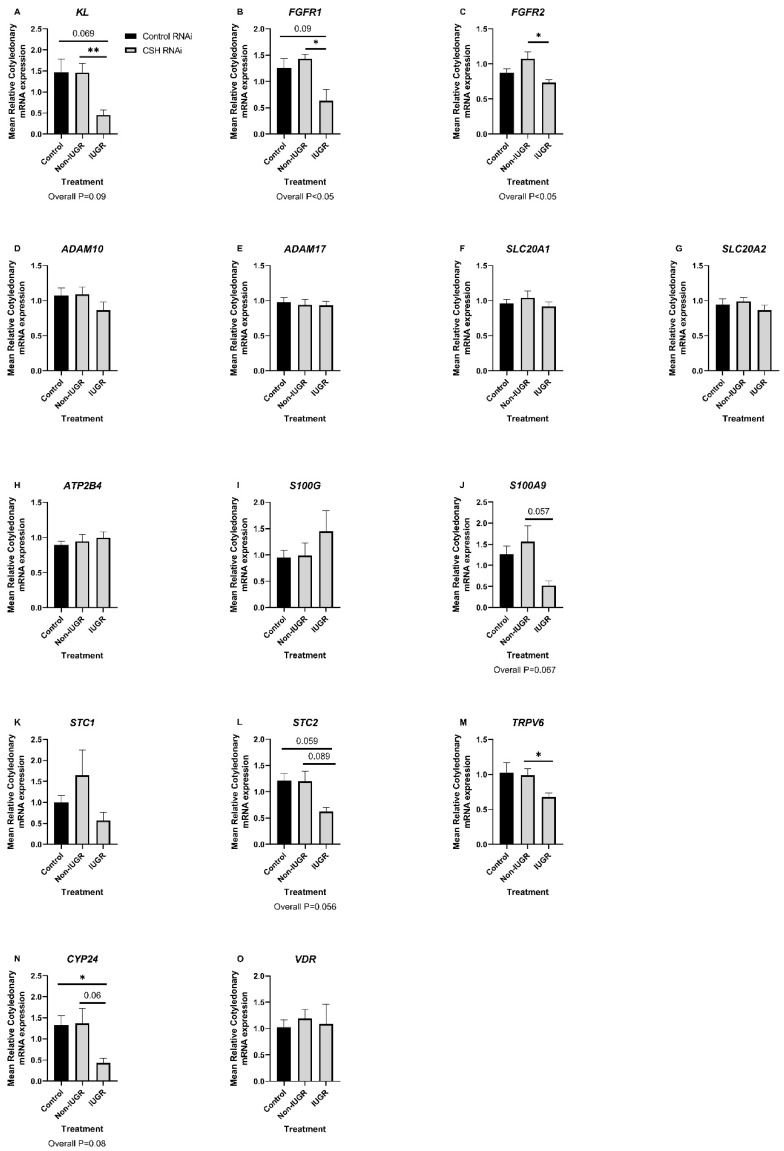
Relative expression of candidate mRNAs with roles in calcium, phosphate, and vitamin D signaling, transport, and metabolism in cotyledons from Control RNAi, CSH RNAi Non-IUGR, and CSH RNAi IUGR pregnancies. Data are presented as means ± SEM. Control RNAi, *n* = 10; CSH RNAi Non-IUGR, *n* = 6; CSH RNAi IUGR, *n* = 4. CSH, chorionic somatomammotropin; IUGR, intrauterine growth restriction; RNAi, RNA interference. * *p* < 0.05, ** *p* < 0.01.

**Table 1 ijms-23-07795-t001:** Effect of CSH RNAi on serum calcium abundance. Data are presented as means ± SEM. Different superscripts indicate statistical significance. (Control RNAi *n* = 10; CSH RNAi Non-IUGR *n* = 6; CSH RNAi IUGR *n* = 4). CSH, chorionic somatomammotropin; IUGR, intrauterine growth restriction; N/A, samples/results not available; RNAi, RNA interference.

		CSH RNAi		Overall *p* Value
	Control RNAi	Non-IUGR	IUGR	
**Uterine**				
Uterine Arterial Calcium (meq/L)	2.694 ± 0.054	2.594 ± 0.026	2.776 ± 0.029	>0.10
Uterine Venous Calcium (meq/L)	2.654 ± 0.052	2.538 ± 0.026	2.718 ± 0.016	0.09
Uterine Artery-Vein Calcium Gradient (meq/L)	0.040 ± 0.007	0.056 ± 0.002	0.058 ± 0.016	>0.10
Uterine Calcium Uptake (meq/min)	0.071 ± 0.012 ^ab^	0.080 ± 0.004 ^a^	0.057 ± 0.010 ^b^	>0.10
Uterine Calcium Uptake per kg of Uterus (meq/min)	0.109 ± 0.023	0.133 ± 0.004	0.112 ± 0.014	>0.10
Uterine Calcium Uptake per kg Fetus (meq/min)	0.019 ± 0.003	0.021 ± 0.001	0.021 ± 0.003	>0.10
Uterine Calcium Uptake per kg Placenta (meq/min)	0.161 ± 0.025	0.209 ± 0.009	0.172 ± 0.032	>0.10
**Umbilical**				
Umbilical Arterial Calcium (meq/L)	2.896 ± 0.042	2.878 ± 0.064	2.899 ± 0.104	>0.10
Umbilical Venous Calcium (meq/L)	2.977 ± 0.042	2.937 ± 0.064	2.984 ± 0.107	>0.10
Umbilical Vein-Artery Gradient (meq/L)	0.081 ± 0.005 ^a^	0.060 ± 0.009 ^b^	0.086 ± 0.004 ^ab^	<0.05
Umbilical Calcium Uptake (meq/min)	0.063 ± 0.006 ^a^	0.048 ± 0.008 ^ab^	0.038 ± 0.008 ^b^	0.089
Umbilical Calcium Uptake per kg of Uterus (meq/min)	0.089 ± 0.011	0.078 ± 0.013	0.075 ± 0.014	>0.10
Umbilical Calcium Uptake per kg Fetus (meq/min)	3.759 ± 0.7171	2.325 ± 0.4301	2.078 ± 0.695	>0.10
Umbilical Calcium Uptake per kg Placenta (meq/min)	0.145 ± 0.016	0.123 ± 0.019	0.105 ± 0.007	>0.10
**Uteroplacental**				
Uteroplacental Calcium Utilization (meq/L)	0.008 ± 0.016	0.032 ± 0.006	0.019 ± 0.012	>0.10

**Table 2 ijms-23-07795-t002:** Effect of CSH RNAi on serum 25(OH)D abundance. Data are presented as means ± SEM. Different superscripts indicate statistical significance. (Control RNAi *n* = 4–10; CSH RNAi Non-IUGR *n* = 6; CSH RNAi IUGR *n* = 4). CSH, chorionic somatomammotropin; IUGR, intrauterine growth restriction; N/A, samples/results not available; RNAi, RNA interference.

		CSH RNAi		Overall *p* Value
	Control RNAi	Non-IUGR	IUGR	
**Uterine**				
Uterine Arterial 25(OH)D (ng/mL)	560.5 ± 43.74 ^a^	N/A	322.9 ± 68.68 ^b^	<0.05
Uterine Venous 25(OH)D (ng/mL)	495.0 ± 51.87 ^a^	485.3 ± 52.62 ^ab^	267.2 ± 28.82 ^b^	<0.05
Uterine Artery-Vein 25(OH)D Gradient (ng/mL)	61.63 ± 26.96	N/A	59.07 ± 28.82	>0.10
Uterine 25(OH)D Uptake (µg/min)	38.77 ± 12.81	N/A	84.47 ± 33.26	>0.10
Uterine 25(OH)D Uptake per kg of Uterus (µg/min)	85.59 ± 36.63	N/A	157.7 ± 58.69	>0.10
Uterine 25(OH)D Uptake per kg Fetus (µg/min)	21.74 ± 12.31	N/A	25.71 ± 9.403	>0.10
Uterine 25(OH)D Uptake per kg Placenta (µg/min)	192.0 ± 113.6	N/A	204.2 ± 87.33	>0.10
**Umbilical**				
Umbilical Arterial 25(OH)D (ng/mL)	639.7 ± 93.39	683.3 ± 127.6	830.3 ± 230.2	>0.10
Umbilical Venous 25(OH)D (ng/mL)	555.4 ± 80.09	573.0 ± 74.25	586.9 ± 170.4	>0.10
Umbilical Vein-Artery 25(OH)D Gradient (ng/mL)	−84.26 ± 71.34	−110.3 ± 132.6	−243.4 ± 104.3	>0.10
Umbilical 25(OH)D Uptake (µg/min)	−63.13 ± 51.85	−63.28 ± 90.15	−227.4 ± 105.2	>0.10
Umbilical 25(OH)D Uptake per kg of Uterus (µg/min)	−99.51 ± 62.51	−118.7 ± 150.5	−468.5 ± 226.8	>0.10
Umbilical 25(OH)D Uptake per kg Fetus (µg/min)	−16.92 ± 13.12	−14.89 ± 22.05	−98.13 ± 55.18	0.09
Umbilical 25(OH)D Uptake per kg Placenta (µg/min)	−162.9 ± 122.0	−165.6 ± 234.2	−723.6 ± 434.9	>0.10
**Uteroplacental**				
Uteroplacental 25(OH)D Utilization (µg/min)	155.3 ± 84.27	N/A	252.9 ± 139.7	>0.10

**Table 3 ijms-23-07795-t003:** Correlations between biometric data and the caruncular and cotyledonary mRNA expression of calcium, phosphate, and vitamin D regulatory molecules.

Biometric Parameter	mRNA	Caruncle		Cotyledon	
		r	*p*-Value	r	*p*-Value
Maternal Weight	*KL*	0.098	0.205	0.083	0.232
Maternal Weight	*FGFR1*	0.001	0.896	0.116	0.142
Maternal Weight	*FGFR2*	0.070	0.261	0.389	0.006
Maternal Weight	*ADAM10*	0.003	0.815	0.158	0.083
Maternal Weight	*ADAM17*	0.021	0.538	0.005	0.783
Maternal Weight	*SLC20A1*	0.060	0.299	0.195	0.059
Maternal Weight	*SLC20A2*	0.072	0.251	0.019	0.586
Maternal Weight	*ATP2B4*	0.007	0.731	0.039	0.433
Maternal Weight	*PTHrP*	0.262	0.025	N/A	N/A
Maternal Weight	*S100G*	0.049	0.346	0.024	0.522
Maternal Weight	*S100A9*	N/A	N/A	0.086	0.223
Maternal Weight	*STC1*	0.233	0.042	0.105	0.205
Maternal Weight	*STC2*	0.059	0.314	0.093	0.203
Maternal Weight	*TRPV6*	0.022	0.535	0.091	0.210
Maternal Weight	*CYP24*	0.050	0.341	0.115	0.168
Maternal Weight	*VDR*	0.014	0.615	0.012	0.645
Uterine Weight	*KL*	0.058	0.338	0.184	0.067
Uterine Weight	*FGFR1*	0.025	0.519	0.009	0.699
Uterine Weight	*FGFR2*	0.002	0.851	0.004	0.803
Uterine Weight	*ADAM10*	0.226	0.034	0.008	0.714
Uterine Weight	*ADAM17*	0.140	0.105	0.001	0.880
Uterine Weight	*SLC20A1*	0.085	0.214	0.001	0.879
Uterine Weight	*SLC20A2*	0.008	0.710	0.127	0.147
Uterine Weight	*ATP2B4*	0.185	0.058	0.045	0.398
Uterine Weight	*PTHrP*	0.177	0.072	N/A	N/A
Uterine Weight	*S100G*	0.0003	0.952	7.535 × 10^−5^	0.972
Uterine Weight	*S100A9*	N/A	N/A	0.060	0.312
Uterine Weight	*STC1*	0.199	0.06	0.058	0.351
Uterine Weight	*STC2*	0.0003	0.942	0.018	0.584
Uterine Weight	*TRPV6*	0.023	0.524	0.094	0.201
Uterine Weight	*CYP24*	0.024	0.511	0.039	0.432
Uterine Weight	*VDR*	0.080	0.226	0.091	0.197
Fetal Weight	*KL*	0.06	0.326	0.222	0.041
Fetal Weight	*FGFR1*	0.055	0.333	0.194	0.05
Fetal Weight	*FGFR2*	0.041	0.394	0.120	0.160
Fetal Weight	*ADAM10*	0.161	0.079	0.076	0.238
Fetal Weight	*ADAM17*	0.005	0.767	0.002	0.854
Fetal Weight	*SLC20A1*	0.241	0.028	0.004	0.806
Fetal Weight	*SLC20A2*	0.167	0.074	0.090	0.227
Fetal Weight	*ATP2B4*	0.09	0.20	0.115	0.170
Fetal Weight	*PTHrP*	0.281	0.02	N/A	N/A
Fetal Weight	*S100G*	0.007	0.725	0.002	0.854
Fetal Weight	*S100A9*	N/A	N/A	0.225	0.04
Fetal Weight	*STC1*	0.130	0.142	0.209	0.065
Fetal Weight	*STC2*	0.002	0.866	0.202	0.053
Fetal Weight	*TRPV6*	4.328 × 10^−5^	0.978	0.002	0.863
Fetal Weight	*CYP24*	0.139	0.105	0.105	0.190
Fetal Weight	*VDR*	5.289 × 10^−6^	0.992	0.224	0.035
Placental Weight	*KL*	0.049	0.376	0.045	0.383
Placental Weight	*FGFR1*	0.015	0.615	0.045	0.369
Placental Weight	*FGFR2*	8.461 × 10^−6^	0.990	0.031	0.486
Placental Weight	*ADAM10*	0.159	0.08	3.854 × 10^−5^	0.979
Placental Weight	*ADAM17*	0.054	0.325	0.002	0.848
Placental Weight	*SLC20A1*	0.288	0.015	0.034	0.451
Placental Weight	*SLC20A2*	0.004	0.789	0.002	0.878
Placental Weight	*ATP2B4*	0.084	0.215	0.068	0.296
Placental Weight	*PTHrP*	0.099	0.190	N/A	N/A
Placental Weight	*S100G*	0.032	0.451	0.014	0.625
Placental Weight	*S100A9*	N/A	N/A	0.022	0.548
Placental Weight	*STC1*	0.032	0.478	0.081	0.270
Placental Weight	*STC2*	0.028	0.492	0.077	0.249
Placental Weight	*TRPV6*	0.128	0.122	0.018	0.585
Placental Weight	*CYP24*	0.01	0.678	0.007	0.736
Placental Weight	*VDR*	0.0006	0.918	0.208	0.043

**Table 4 ijms-23-07795-t004:** Calculations for Nutrient Uptake and Utilization. Abbreviations: UTA = uterine artery, UTV = uterine vein, UMA = umbilical artery, UMV = umbilical vein, [ ] = concentration.

Parameter	Formula
Uterine uptake	uterine blood flow (mL/min) × ([UTA] − [UTV])
Umbilical uptake	umbilical blood flow (mL/min) × ([UMV] − [UMA])
Uteroplacental utilization	uterine uptake–umbilical uptake
Uterine uptake relative to uterine, fetal, or placental weight	uterine uptake/uterine, fetal, or placental weight
Umbilical uptake relative to uterine, fetal, or placental weight	umbilical uptake/uterine, fetal, or placental weight
Uterine artery–uterine vein gradient	[UTA] − [UTV]
Umbilical vein–umbilical artery gradient	[UMV] − [UMA]
Uterine artery-umbilical artery gradient	[UTA] − [UMA]

## Data Availability

All data generated or analyzed during this study are included in this published article and its Appendix A.

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
