# Peer review of "Uptake of Phosphate, Calcium, and Vitamin D by the Pregnant Uterus of Sheep in Late Gestation: Regulation by Chorionic Somatomammotropin Hormone"

_ijms, 2022, doi:10.3390/ijms23147795_

Round 1

Reviewer 1 Report

Manuscript ID.  ijms-1763216

 Uptake of phosphate, calcium, and vitamin D by the pregnant uterus of sheep in late gestation: Regulation by chorionic somatomammotropin hormone.

 This work deals with an interesting topic and the authors present new findings on the relative abundance and utilization of calcium, phosphate and vitamin D in late gestation. Important evidence for the role of CSH in mineral regulation is provided.

However, there are serious problems in the presentation of the results and also in the discussion that make it difficult to understand the work.

The authors need to rewrite the results and the discussion sections according to the results presented and not based on trends.

 Major comments

 ·         Introduction

The authors mention two studies on fetal phenotypes with different consequences, however, it would be interesting to know what determines the condition, whether non-IUGR or IUGR, when does one or the other condition occur?

 ·         In the hypothesis, the authors postulate that CSH RNAi could decrease the utilization of phosphate, calcium and vitamin D utero-placenta. If this occurs, which of the phenotypes are they referring to? Or would both conditions be affected?

 ·         In general, tables 1, 2 and S1 are very confusing, since results are explained that are not found in the table or results that are significant are not explained.

 For example, in Table 1 the authors begin by comparing non-IUGR CSH RNAi and IUGR results, which are not statistically significant and are referred to as a trend. Then the authors make a comparison between IUGR CSH RNAi and control RNAi. However, it is very confusing that the results that are significant are not mentioned while results that have a "trend" are outlined. Moreover, it is completely unclear why different comparisons are made between the control and others between the phenotypes. Thus, the results part should be completely rewritten in a scientific manner that is understandable.

 ·         Results of section 2.2.2 Phosphate: Although it was mentioned in the methodology section that the results and samples for non-IUGR are not available, it should be stated again in the results section when the results are described and the groups are compared.  It was stated that there are no differences between CSH RNAi concentrations. However, since it was not measured in non-IUGR, it should be underlined that there were no differences in the IUGR phenotype, because the values for the non-IUGR group were missing. This is important because otherwise the interpretation is quite misleading.

 ·         Line 270, in the discussion, the authors claim that phosphate signaling in cotyledon and canuncules was affected, but it should be specified that it was only in cotyledon, since in canuncules it was not significant.

This way if dealing with the data is found in the description or interpretation of many results presented in this manuscript. The authors claim that there were changes, but indeed there were no statistically significant differences. Why doing statistics if differences are proposed where no significant changes were found?? This is quite unscientific.

 ·         In the discussion the authors mention that there are no differences in phosphate in utero and umbilical vessels or uptake etc. However, this cannot be seen in the table, indeed it even seems that phosphate concentrations might be decreased. It is (once again!) not clear which statistical analyses were made: between groups or between umbilical or uterine vessels?

 Minor comments

 ·         In result 2.1 the authors describe that RNAi control plasma samples were used. However, in the caption they mention CSH RNAi non-IUGR and CSH RNAi IUGR. If they included the others, where are those results?

·         In Table 1, N/A appears in the legend, however, there is no N/A in the table. Please do not copy and paste the legend of the figures of the tables in all the others.

In table 1 and 2 the values <0.05 are in bold, for what reason?

·         2.2.3 fix punctuation between 25(. OH)D

·         The figures are too small, it is not possible to read the axis.

·         In line 156, it should say calcium and phosphate in caruncule is not altered, since it is not significant.

·         In the discussion on line 265 specify that lower expression of mRNA is found in cotyledon

Author Response

Please see attachment with responses to Reviewer's 1 and 2

Reviewer 2 Report

In the present study, the Authors investigated phosphate, calcium and vitamin D metabolite 25(OH)D concentrations in uterine and umbilical blood vessels. Additionally, they determined the impact of CSH RNAi as well as the expression of candidate mRNAs on the concentration and regulation of phosphate, calcium and 25(OH)D in uterine vessels, caruncles and placental cotyledons in both Non-IUGR and IUGR pregnancies.

The Authors demonstrated that CSH RNAi Non-IUGR pregnancies had a lower umbilical vein – umbilical artery calcium gradient and less cotyledonary and phosphate compared to Control RNAi pregnancies. Moreover, they showed that CSH RNAi IUGR pregnancies had less umbilical calcium uptake, lower uterine arterial and venous concentrations of 25(OH)D, and trends for lower umbilical 25(OH)D uptake compared to Control RNAi pregnancies. Finally they showed that CSH RNAi IUGR pregnancies had decreased umbilical uptake of calcium, less uterine venous 25(OH)D, lower caruncular expression of SLC20A2 mRNA and lower cotyledonary expression of FGFR1 and FGFR2 mRNAs compared to CSH RNAi Non-IUGR pregnancies. The Authors concluded that CSH is actively involved in the regulation of mineral transport and utilization at the ovine maternal-conceptus interface, thus providing further mechanistic understanding of the hormonal regulation of mineral transport during pregnancy in mammalian species.

This is a very interesting study addressing an important issue as hormonal regulation of mineral transport during pregnancy thorough the placenta, uterine and  umbilical blood vessels. Thus, it is of great interest to the readers of the International Journal of Molecular Sciences.
However, there are some points that the Authors must address before publication

1.    Abstract. Please, clarify that caruncles and cotyledons are the maternal and fetal parts of placentome (e.g. feto-maternal interface). Moroever, add that 25(OH)D represent vitamin D metabolite and the meaning SLC20A2, FGFR1 and 2 (first time you mentioned them).

2.   It is clear that the Authors previously published the experimental design but it could be very useful to add a brief description in the methods section (e.g. why you choose day 132 to perform your analyses).

3. The Authors stated that they included in the study 10 Control RNAi; 6 CSH RNAi Non-IUGR, and 4 CSH RNAi IUGR. However, when you analyzed the effects of CSH RNAi on serum calcium (table 1) and 25(OH)D (table 2) abundance you included only 4-9 Control RNAi. Please, clarify

4.    Please, check title legend of table 2. Probably there is a mistake since you reported the effect of CSH RNAi on serum 25(OH)D abundance but you titled it “Effect of CSH RNAi on serum calcium abundance”

5. It is very interesting that when that Authors evaluated the effect of CSH RNAi on calcium and phosphate abundance and on candidate mRNAs with roles in the regulation of calcium, phosphate and vitamin D signaling in cotyledons and caruncles, they showed main effects on fetal part of the placentome rather that maternal part. Can you, please, comment these results?

6.      The Authors should add in the Methods section the list of genes that they analyzed explaining the rationale (e.g. sodium dependent phosphate transporter).

7.      In Figure 2B and 2D, the Authors reported a significant differences in calcium and phosphate levels between controls and non IUGR in cotyledons. However, it is very strange that you did not find significant differences between non-IUGR and IUGR (graphically the difference is greater relative to control-non IUGR). Please, clarify.

8.      Can the Authors, please check the statistic of Figure 3A, 3I, 3K, 3M and 3N? The bars are really different between groups but you did not report significant differences.

9.     It could be very interesting if the Authors could correlate their data with maternal and fetal biometry data maybe adding two tables (Data on maternal and on fetal biometrics).

Author Response

Please see attached file with responses to Reviewer's 1 and 2
